# Multi-trait and multi-environment Bayesian analysis to predict the G x E interaction in flood-irrigated rice

**Antônio Carlos da Silva Júnior**[1]*, **Isabela de Castro Sant'Anna**[2], **Michele Jorge Silva Siqueira**[1], **Cosme Damião Cruz**[1], **Camila Ferreira Azevedo**[3], **Moyses Nascimento**[3], **Plínio César Soares**[4]

1 Departamento de Biologia Geral, Universidade Federal de Viçosa, Viçosa, Minas Gerais, Brasil, 2 Centro de Seringueira e Sistemas Agroflorestais, Instituto Agronômico (IAC), São Paulo, Brasil, 3 Departamento de Estatística, Universidade Federal de Viçosa, Viçosa, Minas Gerais, Brasil, 4 Empresa de Pesquisa Agropecuária de Minas Gerais–EPAMIG, Viçosa, Minas Gerais, Brazil

* antonio.silva.c.junior@gmail.com

## Abstract

The biggest challenge for the reproduction of flood-irrigated rice is to identify superior genotypes that present development of high-yielding varieties with specific grain qualities, resistance to abiotic and biotic stresses in addition to superior adaptation to the target environment. Thus, the objectives of this study were to propose a multi-trait and multi-environment Bayesian model to estimate genetic parameters for the flood-irrigated rice crop. To this end, twenty-five rice genotypes belonging to the flood-irrigated rice breeding program were evaluated. Grain yield and flowering were evaluated in the agricultural year 2017/2018. The experimental design used in all experiments was a randomized block design with three replications. The Markov Chain Monte Carlo algorithm was used to estimate genetic parameters and genetic values. The flowering is highly heritable by the Bayesian credibility interval: $h^2 = 0.039$–0.80, and 0.02–0.91, environment 1 and 2, respectively. The genetic correlation between traits was significantly different from zero in the two environments (environment 1: -0.80 to 0.74; environment 2: -0.82 to 0.86. The relationship of $CV_e$ and $CV_g$ higher for flowering in the reduced model ($CV_g/CV_e = 5.83$ and 13.98, environments 1 and 2, respectively). For the complete model, this trait presented an estimate of the relative variation index of: $CV_e = 4.28$ and 4.21, environments 1 and 2, respectively. In summary, the multi-trait and multi-environment Bayesian model allowed a reliable estimate of the genetic parameter of flood-irrigated rice. Bayesian analyzes provide robust inference of genetic parameters. Therefore, we recommend this model for genetic evaluation of flood-irrigated rice genotypes, and their generalization, in other crops. Precise estimates of genetic parameters bring new perspectives on the application of Bayesian methods to solve modeling problems in the genetic improvement of flood-irrigated rice.

**Data Availability Statement:** The data belong to the Agricultural Research Company of Minas Gerais in Brazil. There are ethical restrictions on

sharing the used data set, because this one still contains important about the genotypes information; in addition, the data are owned by state-owned organization. However, data access requests may be directed to: epamigsudeste@epamig.br.

**Funding:** The authors would like to thank the Research Support Foundation of the State of Minas Gerais, the National Council for Scientific and Technological Development, and the Coordination for the Improvement of Higher Education Personnel for the financial support and research of Embrapa Rice and Beans Dr. Orlando Peixoto de Morais (in memory) and Prof. Dr. Fabyano Fonseca e Silva (in memory). This study was financed in part by the Coordination for the Improvement of Higher Education Personnel - Brazil (CAPES) - Financial Code 001. The authors gratefully acknowledge the Fundação de Amparo à Pesquisa do Estado de São Paulo (FAPESP) for researcher fellowship to ICS 2018/26408-0. The funders had a role in study design, data collection and analysis, the decision to publish, and the preparation of the manuscript.

**Competing interests:** The authors have declared that no competing interests exist.

# Introduction

Rice (*Oryza sativa* L.) is one of the most important crops in the world and is considered one of the main annual crops in Brazil [1]. Rice breeding is primarily aimed at the development of high-yielding varieties with specific grain qualities, resistance to abiotic and biotic stresses in addition to superior adaptation to the target environment [2, 3]. In this case, the breeder needs to realize mutual relationships and knowledge of the interdependence between agronomically important traits that can improve accuracy selection [3].

Specifically, in the rice crop, evaluating multiple traits rather than a single trait aims to maximize grain yield and quality. This is possible through the exploration of genetic correlations between traits. In multi-trait analysis, the prediction of secondary traits can be used to improve the prediction of primary characteristics, especially when they have low heritability. Although consideration of the genetic correlation between traits is essential, modeling interactions between phenotypes provides essential information for developing breeding strategies that cannot be carried out with conventional multivariate approaches alone [2, 4, 5].

Biometric methods available are useful for analyzing a single trait measured in a single environment or across multiple environments with the genotype x environment (G x E) interaction [6–9]. This interaction can be defined as the differential response of genotypes to environmental variation [10], and offers an even greater challenge for the breeder [11]. The information from a network of experiments obtains a multi-trait and multi-environment (MTME) structure but presents limited statistical methodology that not correctly represents genetic and phenotypic variation in the data [12]. Therefore, genetic correlations and G x E interaction require more complex models that are difficult to converge in the context of mixed linear models [11]. Thus, Bayesian inference has become a useful statistical tool for dealing with complex models [13].

Bayesian inference has been used successfully in studies with complex models. [11] evaluated the multi-trait and multi-environment Bayesian model considering the G x E interaction for nitrogen use efficiency components in tropical corn. [13], applied such models through Bayesian inference applied to the breeding of *jatropha curcas* for bioenergy. [14], used these models in the genetic selection of soybean progenies. [15], demonstrate such models in phenotypic and genotypic data in corn and wheat. However, few studies combine models of multiple traits in a multi-environment under a Bayesian point of view, mainly for rice cultivation.

Therefore, the objectives of this study were to propose a multi-trait and multi-environment Bayesian model to estimate genetic parameters for the flood-irrigated rice crop. In addition to comparing: (i) the complete model (considering the interaction between genotypes and environment) with the restricted model (not considering the interaction); (ii) estimates of genetic parameters of models with single and multiple traits, for grain yield and flowering.

# Material and methods

## Description of the experiment

The experiments were carried out in the State of Minas Gerais—Brazil, in the experimental fields of Agricultural Research Corporation of Minas Gerais State (EPAMIG), in the cities of Lambari (21˚ 58' 11.24" S, 45˚ 20' 59.6" W) and Janaúba (15˚ 48' 77" S, 43˚ 17' 59.09" W). Twenty-five genotypes belonging to the flood-irrigated rice breeding program of the Southeast region of the state of Minas Gerais were evaluated, and five of these genotypes were used as experimental controls (Rubelita, Seleta, Ourominas, Predileta, and Rio Grande). These genotypes were evaluated in comparative trials after multiple generations of selection, and in addition, they are known for their high yield, uniform growth rate and plant growth, resistance to

major diseases, and for their excellent grain quality. Grain yield (GY, Kg ha$^{-1}$) and flowering period in days (FL) were evaluated in the agricultural year 2017/2018. The experimental design was a randomized complete block design with three replications. The experiments were conducted in floodplain soils with continuous flood irrigation. Grain production data in grams per useful plot were used, later converted into kilograms per hectare. Management practices were carried out according to recommendations for flood-irrigated rice in the region.

The useful area consisted of 4 m central of three internal rows (4 m x 0.9 m, totaling 3.60 m$^2$). The soil preparation was carried out by plowing and harrowing around 30 days before sowing. For planting fertilization, a mixture of 100 kg ha$^{-1}$ of ammonium sulfate, 300 kg ha$^{-1}$ of simple superphosphate, and 100 kg ha$^{-1}$ of potassium chloride was used, applied in the plot, and incorporated into the soil before planting. The fertilization in the top dressing was carried out approximately 60 days after the installation of the experiments, with 200 kg ha$^{-1}$ of ammonium sulfate. The weeds were controlled with the use of herbicides and manual weeding. Sowing was carried out in the planting line with a density of 300 seeds m$^{-2}$. The irrigation started around 10–15 days after the implantation of the experiments, and the water was only removed close to the maturation of the materials. The harvest was carried out when the grains reached a humidity of 20–22%. Grain production data were obtained by weighing all grains harvested in the useful plot, after cleaning and uniform drying in the sun until they reached a humidity of 13%. Days for flowering correspond to the number of days from sowing to flowering when the plot presented approximately 50% of plants with panicles.

### Biometric analysis

Grain yield (GY) and flowering period in days (FL) were analyzed using single- and multi-trait models using the Bayesian Markov Chain Monte Carlo (MCMC) approach. The objective was to compare: (i) the complete model (considering the interaction between genotypes and environments) with the restricted model (not considering the interaction); (ii) estimates of genetic parameters of models with single and multiple traits, for grain yield and flowering period.

The multi-trait and multi-environment (MTME) model was given by:

$$y = X\beta + W_1 r + W_2 u + \varepsilon$$

which can be rewritten as:

$$\begin{pmatrix} Y_1 \\ \cdots \\ Y_2 \end{pmatrix} = X \begin{pmatrix} \beta(E1,1) \\ \beta(E2,1) \\ \cdots \\ \beta(E1,2) \\ \beta(E2,2) \end{pmatrix} + W_1 \begin{pmatrix} r_1(E1,1) \\ r_2(E2,1) \\ \cdots \\ r_1(E1,2) \\ r_2(E2,2) \end{pmatrix} + W_2 \begin{pmatrix} u(E1,1) \\ u(E2,1) \\ \cdots \\ u(E1,2) \\ u(E2,2) \end{pmatrix} + \begin{pmatrix} \varepsilon(E1,1) \\ \varepsilon(E2,1) \\ \cdots \\ \varepsilon(E1,2) \\ \varepsilon(E2,2) \end{pmatrix},$$

where: y is the vector of the phenotypic values of the two evaluated traits (grain yield, $Y_1$; and flowering period, $Y_2$); X is the incidence matrix of the systematic effects represented by β, assuming $\beta \sim N(\mu_\beta, I_{\otimes}\Sigma_\beta)$ so that *E*1 and *E*2 represent the two environments studied; $W_1$ it is the incidence matrix of the random effect of the environment; r is the ambient effect vector, r $\sim N(0, I_{\otimes}\Sigma_r)$; $W_2$ is the incidence matrix of the effects of the genotype x environment interaction; u is the random effects vector of the genotype x environment interaction, $u \sim N(0, I_{\otimes}\Sigma u)$; and ε is the residual effects vector, $\varepsilon \sim N(0, I_{\otimes}\Sigma_\varepsilon)$.

The (co)variance matrices are given by:

$$\sum_r = \begin{pmatrix} \sigma^2_{ry(1)} & \sigma_{ry(1,2)} & \sigma_{ry,E(1)} & \sigma_{ry,E(1,2)} \\ \sigma_{ry(1,2)} & \sigma^2_{ry(2)} & \sigma_{ry,E(2,1)} & \sigma_{ry,E(2)} \\ \sigma_{ry,E(1)} & \vdots & \sigma^2_{rE(1)} & \cdots \\ \sigma_{ry,E(1,2)} & \cdots & \cdots & \sigma^2_{rE(2)} \end{pmatrix}$$

$$\sum_u = \begin{pmatrix} \sigma^2_{uy(1)} & \sigma_{uy(1,2)} & \sigma_{uy,E(1)} & \sigma_{uy,E(1,2)} \\ \sigma_{uy(1,2)} & \sigma^2_{uy(2)} & \sigma_{uy,E(2,1)} & \sigma_{uy,E(2)} \\ \sigma_{uy,E(1)} & \vdots & \sigma^2_{uE(1)} & \cdots \\ \sigma_{uy,E(1,2)} & \cdots & \cdots & \sigma^2_{uE(2)} \end{pmatrix}$$

$$\sum_\varepsilon = \begin{pmatrix} \sigma^2_{\varepsilon y(1)} & \sigma_{\varepsilon y(1,2)} & \sigma_{\varepsilon y,E(1)} & \sigma_{\varepsilon y,E(1,2)} \\ \sigma_{\varepsilon y(1,2)} & \sigma^2_{\varepsilon y(2)} & \sigma_{\varepsilon y,E(2,1)} & \sigma_{\varepsilon y,E(2)} \\ \sigma_{\varepsilon y,E(1)} & \vdots & \sigma^2_{\varepsilon E(1)} & \cdots \\ \sigma_{\varepsilon y,E(1,2)} & \cdots & \cdots & \sigma^2_{\varepsilon E(2)} \end{pmatrix},$$

where: y represents grain yield and h represents flowering period in days; 1 and 2 represent the two environments studied. The variance-covariance matrices follow an inverted Wishart distribution, which was used as a priori to model the variance-covariance matrix [16].

The package "MCMCglmm" [17] was used. A total of 10,000,000 samples were generated and assuming a flare period and sampling interval of 500,000 and 10 iterations, respectively, this resulted in a final total of 50,000 samples. The convergence of the MCMC was verified by the criterion of [18], performed in two packages R "boa" [19] and "CODA" (convergence diagnosis) [20]. Even though Bayesian and frequentist structures were not directly compared, especially in the field of genetics [21], the same models were also adjusted based on the REML estimation method (Restricted Maximum Likelihood).

The complete models (considering the interaction between genotypes and environments) were compared with the null models (not considering the interaction) by the deviation information criterion (DIC) proposed by [22]:

$$DIC = D(\bar{\theta}) + 2p_D$$

where is a point estimate of the deviation obtained by replacing the parameters with their later means estimates in the probability function and $p_D$ is the effective number of model parameters. Models with a lower DIC should be preferred over models with a higher DIC.

The components of variance, broad-sense heritability, coefficient of variation residual and genetic, variation index, and genotypic correlation coefficients between genetic traits and values were calculated from the posterior distribution. The package "boa" [19] was used to calculate the intervals of greater posterior density (HPD) for all parameters. The later estimates for the broad-sense heritability of grain yield and flowering period in days for each interaction were calculated from the later samples of the variance components obtained by the model,

using the expression:

$$h^{2(i)} = \frac{\sigma_g^{2(i)}}{(\sigma_g^{2(i)} + \sigma_r^{2(i)} + \sigma_\varepsilon^{2(i)})}$$

where: $\sigma_g^{2(i)}$, $\sigma_r^{2(i)}$, and $\sigma_\varepsilon^{2(i)}$ are the genetic, replication, and residual variations for each iteration, respectively.

The genetic correlation coefficients between the pairs of traits in each environment were obtained, as suggested by [23], using the expression below for all models: $\rho_{l(1,2)} = \frac{\sigma_{gl(1,2)}}{\sqrt{\sigma_{gl(1)}^2 \sigma_{gl(2)}^2}}$: genetic correlation between environment and grain yield and $\rho_{h(1,2)} = \frac{\sigma_{gh(1,2)}}{\sqrt{\sigma_{gh(1)}^2 \sigma_{gh(2)}^2}}$: genetic correlation between environment and flowering period in days. All data analysis was conducted using the statistical software package R version 4.1.0.

## Results and discussion

Geweke's convergence criterion indicated convergence for all dispersion parameters by generating 10,000,000 MCMC strings, 500,000 samples for burn-in, and a sampling interval of 10, totaling 50,000 effective samples used for estimating variance components. Similar posterior mean, median and modal estimates were obtained for variance components, suggesting normal-appearing density. However, all chains reached convergence by this criterion. According to the deviation information criteria (DIC), there was positive evidence of interactions between genotypes and environments for all analyzed models (Table 1). However, the DIC values were lower when using the complete model (considering the effects of genotype x environment interaction), in which the difference in relation to the complete model was greater than 2 (Table 1), which according to [22] the use of full model can lead to greater precision in estimating parameters (Table 1). Therefore, since the obtained DIC values were greater than two, it is possible to indicate the superiority of the complete model over the restricted models. On the other hand, as this component of the model is important, the "best" genotypes measured in different environments cannot be the same. However, convergence was not achieved by the AI (Average Information) and EM (Expectation-Maximization) algorithms.

The posterior inferences for mean and higher posterior density range (HPD) considering the multi-trait and multi-environment (MTME) model are described in Table 2. The average values for the grain yield trait varied from 4210.91–3901.56 kg ha$^{-1}$ and flowering in days of 99.40–76.43, in environments 1 and 2, respectively (Table 2). The Bayesian credibility interval (95% probability) for average grain yield corresponds to 4191.70–4227.86 kg ha$^{-1}$, and 3852.60–3946.07 kg ha$^{-1}$ (p<1e-05), in environments 1 and 2, respectively. In relation to the flowering period in days, this interval corresponds to 98.09–100.69 e 74.56–78.22 (p<1e-05), environments 1 and 2, respectively.

**Table 1. Deviation information criteria for the full (considering the G x E interaction) and null (not considering the interaction) models.**

| | | Deviance information criteria (DIC) | |
|---|---|---|---|
| **Model** | **Trait** | **Full Model** | **Null Model** |
| Mult-trait | GY, FL | -308.83 | 1967.79 |
| Single-trait | GY | 1867.49 | 1868.28 |
| Single-trait | FL | 162.19 | 697.67 |

GY: Grain Yield; FL: Flowering Period.

**Table 2. Posterior inferences for mean and highest posterior density range (HPD) considering the proposed complete multi-trait multi-environment model.**

| Trait | EN | post.mean | HPD 95% | |
| | | | LOWER | UPPER |
|---|---|---|---|---|
| GY | 1 | 4210.91*** | 4191.7 | 4227.86 |
| FL | 1 | 99.4*** | 98.09 | 100.69 |
| GY | 2 | 3901.56*** | 3852.6 | 3946.07 |
| FL | 2 | 76.43*** | 74.56 | 78.22 |

*** Significância estatística: p $\leq$ 0.001. GY: Grain Yield; FL: Flowering Period; EN: environment.

Table 3 presents the subsequent inferences for mean and genetic variance; mode, mean, median, and highest posterior density range (HPD) of heritability in the broad sense; and the mode, mean, median, and greater posterior density interval (HPD) of the genetic correlation, considering MTME. The grain yield trait in environment 2 was considered weakly heritable with Bayesian credibility interval (95% probability): $h^2$ = 4.36E-07–9.21E-06, and in environment 1, it was moderately heritable $h^2$ = 0.39–0.757. The low estimate of heritability observed does not depend on the number of samples evaluated, since the Bayesian structure used is essentially recommended for situations involving small sample sizes. In addition, the grain yield trait is largely influenced by the environment as it is a quantitative character [24], which reflects this low estimate of heritability. In relation to flowering period, it is highly heritable by the Bayesian credibility interval (95% probability): $h^2$ = 0.039–0.80, and 0.02–0.91, environment 1 and 2, respectively. And the posterior mean of the genetic correlation between traits was significantly different from zero (95% credibility intervals) in the two environments (environment 1: -0.80 to 0.74; environment 2: -0.82 to 0.86) (Table 3).

Bayesian methods access the posterior density ranges of genetic parameters (Fig 1). The genetic parameters of the flood-irrigated rice genotypes for each trait and their larger posterior density ranges (HPD) were obtained to assist in the selection of genotypes.

In Fig 2, there is a unimodal distribution in which there is a mixture of two distinct populations for posterior density in relation to the genotypic correlation between traits for the complete MTME model. The red line represents the posterior density for environment 1, while the blue line represents the posterior density for environment 2.

**Table 3. Resume of inferences for mean and genetic variance; mode, mean, median, and highest posterior density range (HPD) of heritability in the broad sense; and the mode, mean, median, and highest posterior density range (HPD) of the genetic correlation, considering the complete model multi-trait and multi-environment.**

| Trait | EN | $h^2$ | | | HPD 95% | |
| | | Mode | Mean | Median | LOWER | UPPER |
|---|---|---|---|---|---|---|
| GY | 1 | 0.11 | 0.28 | 0.18 | 0.39 | 0.757 |
| GY | 2 | 1.30E-06 | 3.32E-06 | 2.24E-06 | 4.36E-07 | 9.21E-06 |
| FL | 1 | 0.12 | 0.31 | 0.24 | 0.039 | 0.80 |
| FL | 2 | 0.06 | 0.27 | 0.13 | 0.02 | 0.91 |
| | | Genotypic Correlation | | | HPD 95% | |
| | | Mode | Mean | Median | LOWER | UPPER |
| GY, FL | 1 | -0.0076 | -0.027 | -0.028 | -0.80 | 0.74 |
| GY, FL | 2 | 0.037 | 0.018 | 0.019 | -0.82 | 0.86 |

EN: environment; GY: Grain Yield; FL: Flowering Period; $h^2$: heritability.

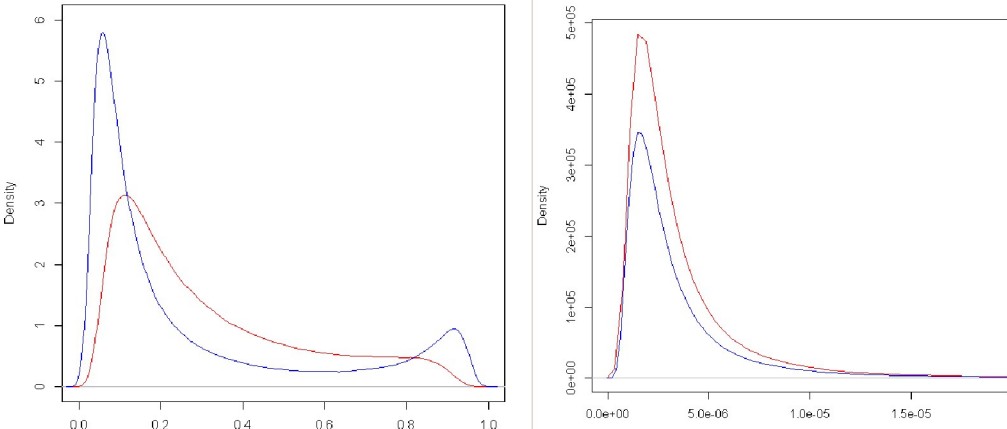

**Fig 1. Posterior density for the complete model proposed by multi-traits multi-environment (left: flowering period and right: grain yield).** The red line represents the posterior density for environment 1, while the blue line represents the posterior density for environment 2.

## Variance estimate

The a posteriori estimates of the genotypic and residual variances for the reduced model (MTME) were very discrepant among the environments (Table 4). The grain yield trait in the

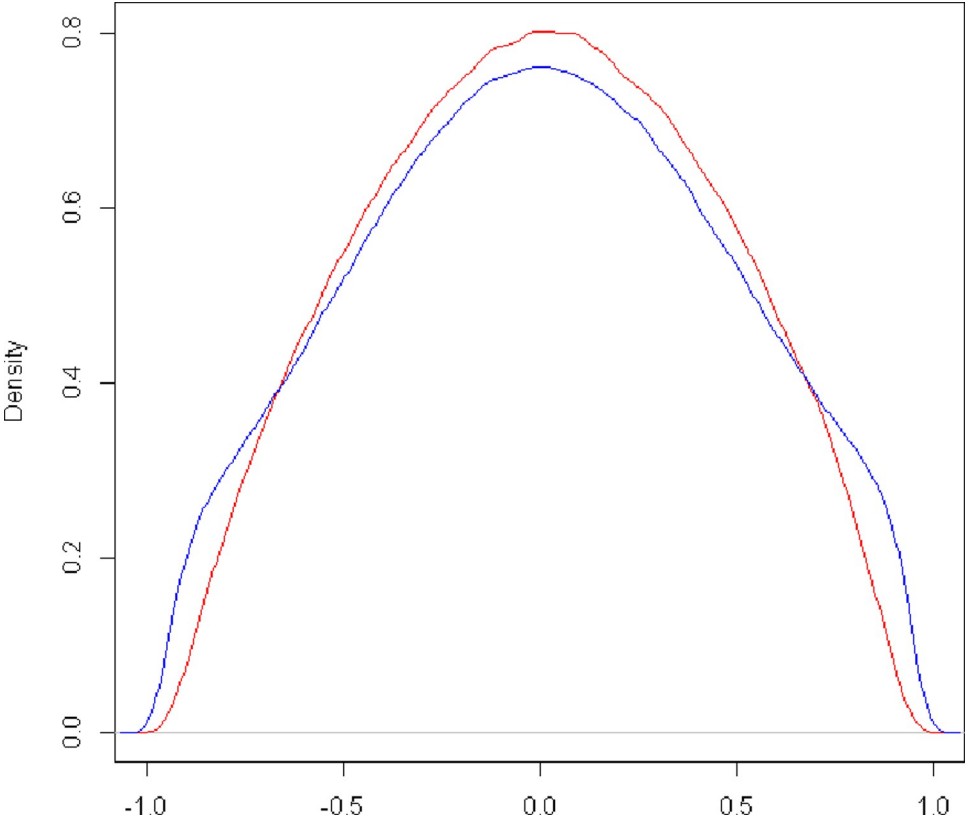

**Fig 2. Posterior density for the genotypic correlation between the grain yield trait and flowering period in days for the model proposed by multi-traits and multi-environment.** The red line represents the posterior density for environment 1, while the blue line represents the posterior density for environment 2.

**Table 4. Genetic parameters for traits grain yield and flowering period in days, in two environments, using multi-trait multi-environment (MTME) models.**

| | Model | Trait | EM | Component | | |
|---|---|---|---|---|---|---|
| | | | | $\sigma^2_g$ | $\sigma^2_r$ | $\sigma^2_{int}$ |
| Multi trait | Null | GY | 1 | 38.93 | 8.86 | - |
| | | GY | 2 | 23.19 | 6.13 | - |
| | | FL | 1 | 14.95 | 4.45E-5 | - |
| | | FL | 2 | 32.63 | 0.19 | - |
| | Full | GY | 1 | 2.63 | 7.24 | 4.64 |
| | | GY | 2 | 2.70 | 7.39 | 4.77 |
| | | FL | 1 | 3.35 | 0.18 | 7.57 |
| | | FL | 2 | 5.69 | 0.34 | 15.91 |

EN: environment; GY: Grain Yield; FL: Flowering Period; $\sigma^2_g$, $\sigma^2_r$, $\sigma^2_{int}$: are the genetic, replication, and interaction variations, respectively.

reduced model obtained a greater estimate of genotypic variance in environment 1, corresponding to a difference of approximately 60% greater, in relation to environment 2. This indicates a greater influence of genetic components on environmental components in the expression of traits. On the other hand, this estimate for the complete model showed less variation between environments, especially for the flowering period in days trait. This result showed the best consistency for the complete model (MTME). Estimates of the variance of the greatest interaction were observed for the flowering period trait.

**Relative variation index.** The ratio of the coefficient of variation genotypic and the coefficient of variation residual ($CV_g/CV_e$) corresponded to the relative variation index. When this index is greater than one unit, it suggests that genetic variation is more influential than residual variation. This was observed in this study for both traits, in the reduced model proposed by MTME (Table 5). For the complete model in the grain yield trait, this relationship was less than one unit. The relationship of $CV_e$ and $CV_g$ higher for flowering period trait in the reduced model ($CV_g/CV_e$ = 5.83 and 13.98, environments 1 and 2, respectively). This trait has greater variability and is highly promising for selection. This is due to its complex genetic inheritance resulting from the involvement of several genes with little effect on the phenotype [24, 25]. For the complete model, this trace presented an estimate of the relative variation index of: $CV_g/CV_e$ = 4.28 and 4.21, environments 1 and 2, respectively (Table 5). Then, this trait presented a higher posterior density interval (HPD) for 80 and 91% heritability, environment 1 and 2,

**Table 5. Coefficient of variation residual ($CV_e$, %), coefficient of variation genotypic ($CV_g$, %) and relative variation index ($CV_g/CV_e$) for the multi-trait and multi-environment model.**

| Model | Trait | EN | $CV_g$(%) | $CV_e$(%) | $CV_g/CV_e$ |
|---|---|---|---|---|---|
| Null | GY | 1 | 0.149 | 0.071 | 2.11 |
| | GY | 2 | 0.122 | 0.063 | 1.92 |
| | FL | 1 | 3.89 | 0.67 | 5.83 |
| | FL | 2 | 7.47 | 0.57 | 13.98 |
| Full | GY | 1 | 0.039 | 0.064 | 0.61 |
| | GY | 2 | 0.042 | 0.070 | 0.60 |
| | FL | 1 | 1.84 | 0.43 | 4.28 |
| | FL | 2 | 3.20 | 0.76 | 4.21 |

EN: environment; GY: Grain Yield; FL: Flowering Period; EN: environment.

respectively, reinforcing the possibility of genetic gain (Table 3). This implies that the variability observed in these traits has genetic predominance, which is interesting in the process of genetic gain in a flood-irrigated rice breeding program. The coefficient of variation genotypic for the grain yield trait was low in both environments. This is justified by the fact that these genotypes belong to advanced comparative trials in which they have gone through several generations of selection.

One of the ways to succeed in breeding programs is related to the accurate prediction of genotypic values, which is closely related to the adoption of adequate models. Thus, in this study, we apply a new statistical approach for estimating variance components in floodplain rice breeding schemes. The implementation of multi-trait multi-environment models Bayesian is straight forward and currently has been widely used due to the possibility of considering a priori knowledge in modeling. In addition to its application wide application in animal breeding [26, 27], Bayesian multi-trait analysis has been reported in plant breeding [11, 13–15, 28].

Bayesian inference has been used since 1986 [29] and has been further explored in recent years [2, 11, 13–15, 30, 31] due to major computational advances and new applications and methodologies [32]. However, Bayesian analysis is based on knowledge of the posterior distribution of the parameters to be estimated. This allows the construction of exact credibility intervals for estimates of random variables and variance components [33]. Values for the 95% distribution credibility interval for the broad-sense heritability parameter found in this study (Table 3) were also presented in the study by [11] to estimate genetic parameters for efficiency of uptake and efficiency of use of N under contrasting soil N levels via MTME models. Another study, based on the estimation of genetic parameters for genetic selection of segregating soybean progenies using the MTME model [14]. The difference between mean, mode, and median of the broad-sense heritability estimates (Table 3) reflects some lack of symmetry in the posterior distribution estimates. The lack of symmetry between mean, mode, and median heritability estimates in posterior distribution estimates was reported by [11, 30].

The low broad-sense heritability observed in the traits does not depend on the number of samples evaluated, since the Bayesian structure used is essentially recommended for situations involving small samples. On the other hand, quantitative traits of agronomic interest, determined by several genes, demonstrate low expression and significantly influenced by the environment [24], reflected in the traits grain yield and flowering period in days.

Based on the results of heritability estimation in the broad sense for the GY and FL traits varied in: $h^2 = 0.39$–$0.757$ and $h^2 = 4.36E{-}07$–$9.21E{-}06$; $h^2 = 0.039$–$0.80$, and $0.02$–$0.91$, environment 1 and environment 2, respectively (Table 3). [34] found heritability estimates of 0.48 and 0.94 for GY and FL, respectively, using 198 rice progenies by the ANOVA technique. [35] evaluated upland rice genotypes, by this same technique, and obtained an estimate of $h^2$ for GY and FL traits 0.35 and 0.77, respectively. [36] obtained results of $h^2$ ranging from 0.44–0.87 for GY, 0.46–0.94 for FL. [37] applied a mixed model in studies using FL and GY traits, found an estimate of $h^2$ de 0,88, and 0,71, respectively. [38], using the ANOVA method, found $h^2$ of 0.76 for FL and [39] estimate of 0.30 for GY. Regarding the estimate of CVs using the ANOVA method for the FL and GY traits, representing 2.98% and 15.28%, respectively [34].

The amounts of data that breeding programs around the world are generating continue to increase; consequently, there is a growing need to extract more knowledge from the data being produced. For this, MTME models are commonly used to take advantage of correlated traces to improve parameter estimation and prediction accuracy. However, when there are a large number of features, implementing these types of models is a challenge. Therefore, it is necessary to develop efficient models of multiple trait and multiple environments for selection, in order to take advantage of multiple correlated features. In this work, was proposed an alternative method to analyze MTME model data that could be useful for genotype selection, and

estimation of genetic parameters in flooded rice in the context of an abundance of traits and environments.

Full-model multi-trait analysis tends to be powerful and provide more accurate estimates than single-trait analysis because the previous method can take into account the underlying correlation structure found in a multi-trait dataset. However, Bayesian and non-Bayesian inferences from the MTME model analysis are complex and computationally demanding [40]. [28] argue that Bayesian multi-trait analysis is more appropriate than ANOVA to perform analyzes and select superior genotypes for genetic improvement since the Bayesian model can capture small genetic differences between families, while ANOVA cannot. In this study, was explained how to make Bayesian inference using a multi-trait multi-environment Bayesian model in plant breeding. These results showed that the approach was efficient in estimating genetic parameters in flooded rice.

The correlation study revealed favorable associations for the traits in studies in two settings. This result indicates that the selection of genotypes characterized by longer flowering period favors grain yield in flood-irrigated rice, which is desirable for rice cultivation since later plants tend to be more productive. However, late cycle cultivars tend to be more productive in relation to the early cycle, since they obtain an increase in the amount of photoassimilates that are translocated to grains [3]. This result justifies the significant correlation between the traits. According to [24], correlations between traits may be the result of pleiotropy or genetic linkage. Thus, if the undesirable correlations are caused by genetic linkage, these associations can be broken by recombination caused by crossing or self-fertilization; consequently, these factors do not necessarily become major impediments to breeding programs [41–43].

Another point that we would like to highlight is that our proposed model is of MTME, but with the restriction that an identity matrix is assumed for the variance-covariance matrix of the environments. However, even with this restrictive assumption in the variance-covariance matrix of the environments, the model has the advantage of taking into account the terms of interaction x trait, genotype x trait, and the environment x genotype x trait. Furthermore, it takes into account the correlated traits. The program of irrigated rice program aims to obtain the desired results in a short period and with precision, therefore, the choice of the model will be use as breeding strategies.

## Conclusion

The multi-trait and multi-environment Bayesian model were efficient to estimate genetic parameters for the flood-irrigated rice crop.

The estimates of genetic parameters bring new perspectives on the application of Bayesian methods to solve modeling problems in the genetic improvement of flood-irrigated rice.

## Author Contributions

**Conceptualization:** Antônio Carlos da Silva Júnior, Isabela de Castro Sant'Anna, Michele Jorge Silva Siqueira, Cosme Damião Cruz, Camila Ferreira Azevedo, Moyses Nascimento.

**Data curation:** Antônio Carlos da Silva Júnior, Isabela de Castro Sant'Anna, Michele Jorge Silva Siqueira, Cosme Damião Cruz, Plínio César Soares.

**Formal analysis:** Antônio Carlos da Silva Júnior, Isabela de Castro Sant'Anna, Michele Jorge Silva Siqueira, Cosme Damião Cruz, Camila Ferreira Azevedo, Moyses Nascimento, Plínio César Soares.

**Funding acquisition:** Antônio Carlos da Silva Júnior, Cosme Damião Cruz, Plínio César Soares.

**Investigation:** Antônio Carlos da Silva Júnior, Cosme Damião Cruz, Camila Ferreira Azevedo, Plínio César Soares.

**Methodology:** Antônio Carlos da Silva Júnior, Isabela de Castro Sant'Anna, Michele Jorge Silva Siqueira, Cosme Damião Cruz, Camila Ferreira Azevedo, Moyses Nascimento.

**Project administration:** Cosme Damião Cruz, Plínio César Soares.

**Resources:** Antônio Carlos da Silva Júnior, Cosme Damião Cruz, Plínio César Soares.

**Software:** Antônio Carlos da Silva Júnior, Isabela de Castro Sant'Anna, Michele Jorge Silva Siqueira, Cosme Damião Cruz, Camila Ferreira Azevedo, Moyses Nascimento.

**Supervision:** Antônio Carlos da Silva Júnior, Michele Jorge Silva Siqueira, Cosme Damião Cruz, Camila Ferreira Azevedo, Moyses Nascimento.

**Validation:** Antônio Carlos da Silva Júnior, Isabela de Castro Sant'Anna, Michele Jorge Silva Siqueira, Cosme Damião Cruz, Moyses Nascimento.

**Visualization:** Antônio Carlos da Silva Júnior, Isabela de Castro Sant'Anna, Michele Jorge Silva Siqueira, Cosme Damião Cruz, Camila Ferreira Azevedo, Moyses Nascimento.

**Writing – original draft:** Antônio Carlos da Silva Júnior, Isabela de Castro Sant'Anna, Michele Jorge Silva Siqueira, Cosme Damião Cruz, Camila Ferreira Azevedo, Moyses Nascimento, Plínio César Soares.

**Writing – review & editing:** Antônio Carlos da Silva Júnior, Isabela de Castro Sant'Anna, Michele Jorge Silva Siqueira, Cosme Damião Cruz, Camila Ferreira Azevedo, Moyses Nascimento, Plínio César Soares.

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
