## [Decision Letter · Decision Letter 0]

28 Dec 2021

PONE-D-21-33197Multi-trait and multi-environment Bayesian analysis to predict the G x E interaction in flood-irrigated ricePLOS ONE

Dear Dr. da Silva Júnior,

Thank you for submitting your manuscript to PLOS ONE. After careful consideration, we feel that it has merit but does not fully meet PLOS ONE’s publication criteria as it currently stands. Therefore, we invite you to submit a revised version of the manuscript that addresses the points raised during the review process.

We look forward to receiving your revised manuscript.

Kind regards,

Mehdi Rahimi, Ph.D.

Academic Editor

PLOS ONE

Journal Requirements:

YES - Specify the role(s) played.

NO authors have competing interests

Reviewers' comments:

Reviewer's Responses to Questions

**Comments to the Author**

1. Is the manuscript technically sound, and do the data support the conclusions?

Reviewer #1: Yes

Reviewer #2: Partly

2. Has the statistical analysis been performed appropriately and rigorously? 

Reviewer #1: Yes

Reviewer #2: No

3. Have the authors made all data underlying the findings in their manuscript fully available?

Reviewer #1: Yes

Reviewer #2: No

4. Is the manuscript presented in an intelligible fashion and written in standard English?

Reviewer #1: Yes

Reviewer #2: Yes

5. Review Comments to the Author

Reviewer #1: The Ms "Multi-trait and multi-environment Bayesian analysis to predict the G x E interaction in flood-irrigated rice" is well written. however discussion must be improved further before acceptance.

LINE 310 should be rewritten.

line 320, correct demonstre to demonstrate

Authors should make uniformity in writing the citation at the end of sentences.

Results and discussion should be written in single section.

Reviewer #2: You have chosen a very good statistical tool i.e. Bayesian analysis to predict the G x E interaction. But the number of environments and traits considered for the study is insufficient. A minimum of three environments and all the important agro-morphological traits affecting the performance of rice crop in flood-irrigated situations should have been taken into consideration for drawing the logical conclusion so that other researchers could have been benefitted from this piece of study.

6. PLOS authors have the option to publish the peer review history of their article (what does this mean?). If published, this will include your full peer review and any attached files.

Reviewer #1: No

Reviewer #2: **Yes: **Jai Prakash Jaiswal

---

## [Author Response · Author response to Decision Letter 0]

4 Mar 2022

Reviewer 1 

The Ms "Multi-trait and multi-environment Bayesian analysis to predict the G x E interaction in flood-irrigated rice" is well written. however discussion must be improved further before acceptance.

And.: We improved the discussion

LINE 310 should be rewritten.

And.: We rewrote this line. 

line 320, correct demonstre to demonstrate

And.: We have corrected this detail, please see L350.

Authors should make uniformity in writing the citation at the end of sentences.

And.: We have corrected this detail. 

Results and discussion should be written in single section.

And.: We wrote in single section, please see. 

Reviewer 2

You have chosen a very good statistical tool i.e. Bayesian analysis to predict the G x E interaction. But the number of environments and traits considered for the study is insufficient. A minimum of three environments and all the important agro-morphological traits affecting the performance of rice crop in flood-irrigated situations should have been taken into consideration for drawing the logical conclusion so that other researchers could have been benefitted from this piece of study.

And.: Thanks for the consideration. The ideal would be 3 or more environments, but we didn't have the resources to create more than 3 environments. One of the main purposes of statistics is to make inferences about the parameters of a model. In the frequentist approach, the unknown parameters are considered fixed and the entire analysis is based on the information contained in the data sample, that is, inferential distributions are assumed for the parameter estimators, and not for the parameters themselves.

Bayesian inference treats the unknown parameter vector (θ) as random quantities, and any initial information about them can be represented using probability distributions, which are called a priori distributions. Thus, such an approach allows incorporating some knowledge about these parameters before the data have been collected. These distributions can be obtained through previous analyses, the researcher's experience in the area in question, or literature reviews on the subject to be addressed.

In summary, to perform Bayesian inference it is necessary to assume an a priori probability distribution (P(θ)) for the parameters to be estimated; and distribution for the sample data called the likelihood function P((θ|Y)).

---

## [Decision Letter · Decision Letter 1]

18 Apr 2022

Multi-trait and multi-environment Bayesian analysis to predict the G x E interaction in flood-irrigated rice

PONE-D-21-33197R1

Dear Dr. da Silva Júnior,

We’re pleased to inform you that your manuscript has been judged scientifically suitable for publication and will be formally accepted for publication once it meets all outstanding technical requirements.

Kind regards,

Mehdi Rahimi, Ph.D.

Academic Editor

PLOS ONE

Additional Editor Comments (optional):

Reviewers' comments:

Reviewer's Responses to Questions

**Comments to the Author**

1. If the authors have adequately addressed your comments raised in a previous round of review and you feel that this manuscript is now acceptable for publication, you may indicate that here to bypass the “Comments to the Author” section, enter your conflict of interest statement in the “Confidential to Editor” section, and submit your "Accept" recommendation.

Reviewer #2: All comments have been addressed

2. Is the manuscript technically sound, and do the data support the conclusions?

Reviewer #2: Yes

3. Has the statistical analysis been performed appropriately and rigorously? 

Reviewer #2: Yes

4. Have the authors made all data underlying the findings in their manuscript fully available?

Reviewer #2: Yes

5. Is the manuscript presented in an intelligible fashion and written in standard English?

Reviewer #2: Yes

6. Review Comments to the Author

Reviewer #2: (No Response)

7. PLOS authors have the option to publish the peer review history of their article (what does this mean?). If published, this will include your full peer review and any attached files.

Reviewer #2: **Yes: **Jai Prakash Jaiswal

---

## [Editor Report · Acceptance letter]

22 Apr 2022

PONE-D-21-33197R1 

Multi-trait and multi-environment Bayesian analysis to predict the G x E interaction in flood-irrigated rice 

Dear Dr. da Silva Júnior:

I'm pleased to inform you that your manuscript has been deemed suitable for publication in PLOS ONE. Congratulations! Your manuscript is now with our production department. 

Kind regards, 

on behalf of

Dr. Mehdi Rahimi 

Academic Editor

PLOS ONE